# The experiences of UK-Chinese individuals during the COVID-19 pandemic: A qualitative interview study

**Mohammed Al-Talib**[1,2]*, **Pippa K. Bailey**[1,2], **Qiaoling Zhou**[2], **Katie Wong**[1]

1 Bristol Medical School, Population Health Sciences, University of Bristol, Bristol, United Kingdom,
2 Southmead Hospital, North Bristol NHS Trust, Bristol, United Kingdom

* mohammed.al-talib@bristol.ac.uk

## Abstract

Infectious disease outbreaks have historically been associated with stigmatisation towards minority groups, specifically those associated with the geographical region that the disease was first identified. We aimed to investigate how the emerging COVID-19 pandemic was experienced by UK-resident individuals of Chinese ethnicity: how their perceived cultural and ethnic identity influenced their experiences, and how early insights into the pandemic in China influenced attitudes and behaviours. We undertook in-depth semi-structured interviews with individuals who self-identified as UK-Chinese. Participants were recruited from three cities in the UK. Interviews were undertaken over the telephone between 9th April 2020 and 16th July 2020. Interviews were digitally recorded and transcribed verbatim. Transcripts were coded using NVivo software and analysed using inductive thematic analysis. Sixteen individuals were interviewed. Three main themes were identified: (1) Attribution of stigma, (2) Pandemic legacies, and (3) Individual versus societal responses. These reflected six sub-themes: (1) Stigmatisation through (mis)identity, (2) Markers of pandemic awareness, (3) Legacies of previous pandemics, (4) Ascription of blame, (5) Extent of freedom, and (6) Implicit faith in government. Experiences of xenophobia included accounts of physical violence. UK-Chinese individuals experienced and perceived widespread xenophobia, in the context of media representations that ascribed blame and exacerbated stigmatisation. Prior experience of respiratory epidemics, and insight into the governmental and societal response in China, contributed to the early adoption of face masks. This in turn marked UK-Chinese individuals as targets for abuse. Awareness is needed to safeguard stigmatized groups from social and economic harm in future infectious disease pandemics.

## Introduction

The COVID-19 pandemic, caused by the SARS-CoV-2 virus, emerged in Wuhan, Hubei Province, China at the end of 2019, with a 76 -day lockdown imposed on 23rd January 2020. The first recorded case of COVID-19 in the UK was confirmed on 29th January 2020, with the first national lockdown coming into force on 26th March 2020. In contrast, the majority of Hubei

**Data Availability Statement:** Anonymised transcripts of the interviews have been uploaded to the University of Bristol's Research Data Repository: https://data.bris.ac.uk/data/; (DOI 10.5523/bris.2cgaw1go9gtyo1z163m822wscs). One participant did not consent to data sharing, so their

interview transcript has not been uploaded. Audio files of the recorded interviews are not suitable for sharing as they carry a high risk of allowing the research participant to be identified, and the content of interviews includes sensitive information. Individuals who wish to access the dataset can contact the researchers directly or search the University of Bristol's Research Data Repository. The dataset name is 'Interviews with UK Chinese Individuals' and the creator is Pippa Bailey. Although the qualitative transcripts have been anonymised, as personal and sensitive issues have been discussed we cannot rule out the risk of identification, and therefore access to these transcripts is controlled. Individual researchers will need to request access to the controlled data through the University of Bristol via the Data Access Committee (DAC) for approval, before data can be shared after their host institution has signed a Data Access Agreement. The procedure for accessing data can be found here: https://www.bristol.ac.uk/staff/researchers/data/accessing-research-data/.

**Funding:** MA-T is funded by an NIHR Academic Clinical Fellowship (ACF-2020-25-006) https://www.nihr.ac.uk. PKB is funded by a Wellcome Trust Clinical Research Career Development Fellowship (214554/Z/18/Z) https://wellcome.org. This research was supported by the Elizabeth Blackwell Institute, University of Bristol, and funded by the Wellcome Trust ISSF3 grant 204813/Z/16/Z. https://www.bristol.ac.uk/blackwell. The funders had no role in study design, data collection and analysis, decision to publish, or preparation of the manuscript. The views expressed in this publication are those of the authors and not necessarily those of NIHR, NHS, the Wellcome Trust, or the UK Department of Health and Social Care.

**Competing interests:** The authors have declared that no competing interests exist.

province emerged from lockdown restrictions on 24th March 2020 [1]. There are 393,141 people of Chinese ethnicity domiciled in the UK, accounting for 0·7% of the UK population [2]. This number increased 59% between 2001 and 2011. Through access to Chinese media sources, as well as contact with friends and family in East Asia, UK-Chinese people could more effectively gain information on and insight into the response of the Chinese/Hong Kongese governments and societies to the pandemic. It is not known how this early information and insight, prior to cases developing in the UK, informed attitudes and influenced behaviours when the pandemic reached the UK.

Previous infectious disease epidemics have been associated with discrimination and xenophobia directed towards minority groups perceived as being associated with the source of infection. During the Severe Acute Respiratory Syndrome (SARS; caused by SARS-CoV-1 virus) pandemic in 2003, Asian-Americans in the USA and Canada reported experiencing stigmatisation, discrimination and racism [3–5]. Multiple cross-sectional surveys have found that individuals of East-Asian descent living in the USA [6–9] and Australia [10] experienced discrimination and xenophobia during the COVID-19 pandemic. However, qualitative research providing in-depth insight into the experiences of individuals of Chinese ethnicity, living in the UK or elsewhere outside East Asia, is lacking, both with respect to the COVID-19 and SARS pandemics.

While the effectiveness of face masks in preventing community SARS-CoV-2 transmission has emerged more recently [11, 12], face masks were readily adopted in East Asia early during the pandemic, at a time when evidence of their effectiveness was lacking. Telephone surveys conducted in Hong Kong in January 2020 revealed 74.5% of people were regularly using face masks voluntarily, prior to the first reported case of COVID-19 in Hong Kong [13]. A mixed-methods study of 91 students from Hong Kong studying in the UK examined perceptions and practices around face mask use early during the pandemic [14]. This group reported strong belief in the effectiveness of face masks: a belief they did not feel was shared by the majority of the UK population. Participants also reported stress due to perceived stigmatisation related to face mask use. It is unclear whether this belief in the effectiveness of face masks extended to the long-term resident Chinese diaspora, and how experiences of stigmatisation were attributed to external infection control measures (i.e. face masks) rather than perceived ethnicity. How recognition of stigma affected behaviours around face mask use is also unknown.

To our knowledge, the lived experiences of UK-resident individuals of Chinese ethnicity during the COVID-19 pandemic have not been investigated. We undertook this qualitative interview study in the early stages of the pandemic (between April and July 2020) to address this. We aimed to investigate i) how perceived cultural and ethnic identity impacted the experiences of UK-Chinese individuals during the pandemic, ii) how behaviours and practices of UK-Chinese individuals were influenced by insight into the pandemic responses in China/East Asia, and iii) the perspectives of UK-Chinese individuals on the long-term impacts of the pandemic on the UK-Chinese community.

## Materials and methods

### Study design and recruitment of participants

In this qualitative interview study, in-depth semi-structured interviews were conducted with individuals who were aged 18 years and over and self-identified as UK-Chinese or Chinese. Individuals were eligible if they were of Chinese heritage and born in the UK, or of Chinese heritage born outside the UK and had migrated to the UK. Interviews were chosen over focus groups to allow more in-depth understanding of personal experience, and the exploration of sensitive topic matters [15].

We employed a convenience sampling strategy through trusted community leaders. We contacted the chairs of 12 UK-Chinese Associations via email or telephone to inform them about the study and invite them to participate. These chairs were asked to distribute letters to members of the associations inviting them to participate, and to identify other potential participants who were then contacted via letter or email to confirm whether they wished to receive further information about the study. Individuals interested in participating after this initial contact were then sent a participant information sheet. We then utilised a snowball sampling strategy, whereby interviewees provided details of further eligible individuals, who were subsequently invited to participate.

## Data collection

In-depth semi-structured interviews were conducted between 9th April 2020 and 16th July 2020. Interviews were conducted over the telephone due to the COVID-19 pandemic. All participants provided informed verbal consent to participation which was audio-recorded and transcribed. Interviews were conducted by one member of the study team (KW). KW is a hospital doctor and research fellow with formal training in qualitative research methods. She identifies as UK-Chinese and knew two participants as acquaintances prior to the study. Matters relating to the research study had not been discussed between KW and these acquaintances prior to the interviews taking place. A flexible topic guide was developed by the study team (S1 File). This served as a broad guide of topics to discuss: the questions listed were not asked verbatim to participants and evolved as interviews progressed. Participants were asked about their experiences of the developing pandemic, and the experiences of family and friends outside of the UK. Participants were asked about whether and how insight into the development of the pandemic in China influenced their views and behaviours. They were asked about their views on pandemic responses in the UK and China, and their thoughts on the future impact of the pandemic on the UK-Chinese community. The following participant demographics were collected at the time of interview: age, gender, place of birth, educational level, and year of emigration to the UK (if applicable). Information on non-participation rates and reasons for non-participation was not collected.

Interviews were undertaken in English for reproducibility and rigour. Participants were given the option of requesting use of a Cantonese translator. The veracity of the translation was confirmed by KW, who speaks conversational Cantonese. Interviews were digitally audio-recorded, transcribed verbatim, anonymised, and then uploaded to NVivo software for analysis.

## Data analysis

We undertook an inductive thematic analysis as described by Braun and Clarke [16]. The research was informed by a critical realist position, which considers an individual account as constructed, but also accepts it as a description of events and personal experiences that have some basis in reality [17]. All transcripts were coded by MA-T. MA-T is a clinical academic with formal training in qualitative methods, and received guidance from PKB, an experienced qualitative researcher. Transcripts were read at least twice to gain familiarisation with the data. Following familiarisation the entire dataset was coded: coding was inductive and data driven. Initial codes were generated by assigning descriptive labels to interesting features of the data and sections of text. Codes were then collated into potential themes based on shared properties and clusters of meaning within the dataset, keeping the research objectives in mind. A subset of transcripts was inductively and independently coded by two co-authors (KW and QZ). QZ is a UK-based hospital doctor who was born in China. Codes and potential themes were then

discussed to maximise rigour and reliability, to identify areas of discrepancy and to refine the themes. The generated themes were then reviewed in terms of their relation to the coded extracts and entire dataset, until coding refinements were not adding anything substantial. We assessed that the sample delivered sufficient information power, as the information provided in the interviews was rich, highly relevant and included contemporaneous accounts of current lived experiences [18, 19]. Themes were refined and defined. A thematic map of the analysis was created, and the report prepared using data extracts to illustrate each theme.

Ethical approval was granted by the University of Bristol Faculty of Health Sciences Research Ethics Committee (reference number 94685). The Consolidated criteria for reporting qualitative studies (COREQ) [20] was adhered to and is provided as S2 File.

## Findings

Sixteen individuals participated in 14 interviews. Two interviews were undertaken with the facilitation of a Cantonese translator. Two participants (husband and wife) wished to be interviewed together. This interview was conducted on speakerphone with questions asked once, with the interviewees able to provide individual responses or discuss and answer collectively. Interview lengths ranged between 39 to 78 minutes. Participant characteristics are presented in Table 1.

Three main themes were identified: (1) Attribution of stigma, (2) Pandemic legacies, and (3) Individual versus societal responses. These reflected six sub-themes: (1) Stigmatisation through (mis)identity, (2) Markers of pandemic awareness, (3) Legacies of previous pandemics, (4) Ascription of blame, (5) Extent of freedom, and (6) Implicit faith in government. A thematic map demonstrating the relationship between main and sub-themes is presented below (Fig 1).

Findings are now presented under the three main themes with reference to the sub-themes. Themes are illustrated by participant quotes. A table of quotes reflecting each theme is provided as S1 Table.

**Table 1. Participant characteristics.**

| Participant number | City | Age range | Gender (self-reported) | Place of birth | Educational level | Year of emigration to UK |
|---|---|---|---|---|---|---|
| 1[a] | 1 | 18–25 | Female | China | Postgraduate | 2019 |
| 2[a] | 1 | 26–35 | Male | China | Postgraduate | 2000 |
| 3 | 1 | 46–55 | Female | China | Undergraduate | 2001 |
| 4 | 2 | 56–65 | Female | Hong Kong | Undergraduate | 1967 |
| 5 | 2 | 26–35 | Female | UK | Postgraduate | n/a |
| 6 | 2 | 46–55 | Female | China | Secondary | Not disclosed |
| 7 | 2 | 26–35 | Male | UK | Postgraduate | n/a |
| 8 | 3 | 56–65 | Male | Hong Kong | Vocational | 1995 |
| 9 | 3 | 46–55 | Male | Hong Kong | Secondary | 1999 |
| 10 | 3 | 76–85 | Female | Hong Kong | Undergraduate | 1958 |
| 11 | 3 | 76–85 | Female | Hong Kong | Secondary | 1978 |
| 12 | 3 | 66–75 | Female | Hong Kong | Vocational | 1972 |
| 13 | 3 | 46–55 | Female | Mauritius | Secondary | 2014 |
| 14 | 3 | Not disclosed | Not disclosed | Hong Kong | Postgraduate | 2003 |
| 15 | 3 | 76–85 | Female | India | Undergraduate | 1965 |
| 16 | 3 | 66–75 | Male | Hong Kong | Secondary | 1993 |

[a] Participants were interviewed together.

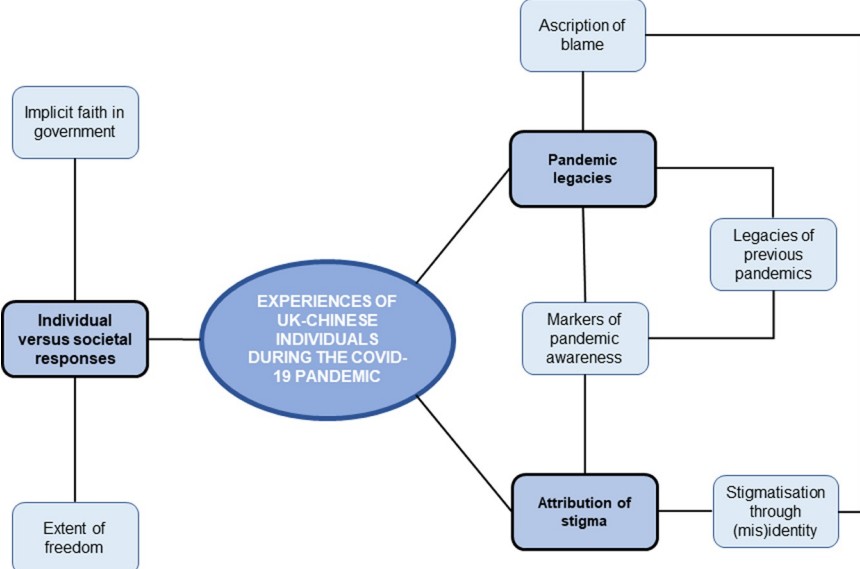

**Fig 1. Thematic schema representing main and sub-themes.** Main themes are indicated in darker boxes, with sub-themes in paler boxes.

## Attribution of stigma

All study participants described experiencing xenophobia either personally, observing it, or described reports of it within their local community. Individuals who had emigrated and those who had been born in the UK had similar experiences. Some participants described explicit acts of violence against individuals within their community.

> *"they get spat on, they get verbal abuse, some of them get stones thrown . . . I report to the police on their behalf and luckily the lockdown reduced the number of cases because nobody goes out." (Participant 8)*

> *"A lot of Chinese students in university they got verbally abused and even some students they got beaten up in the residential hall." (Participant 14)*

**Stigmatisation through (mis)identity.** Beyond these explicit examples of experienced xenophobic abuse, many participants described a heightened awareness in day-to-day life of actions that they perceived as xenophobically motivated.

> *"I got on [a train] and someone started. . . getting their anti-bacteria gel and kind of like gelling their hands as soon as I got on the train. . .I probably think the fact that I was Chinese probably had something to do with it." (Participant 5)*

Even those who had not personally experienced xenophobia felt at risk, or vulnerable to it. Some participants felt that the lockdown, and subsequent reduced time spent outside, had reduced direct instances of xenophobia. Others took active steps to change their behaviours and minimise their perceived risk of being victims of abuse.

*". . .we try to keep a distance with other people because you don't have a close contact with them you won't get a problem. Also we only go out in the daytime. We're not putting ourselves in jeopardy at night." (Participant 15)*

Participants attributed xenophobia, in part, to their being identified by others as ethnically Chinese. One participant described xenophobia experienced by individuals of other ethnic backgrounds due to misidentification as being of Chinese ethnicity.

*"Yeah, the hate crimes, oh you Chinese, you virus, don't come here, go back to China, but the thing is they cannot tell whether they are Chinese or Malaysian or Singapore. . .they just Asian face." (Participant 14)*

**Markers of pandemic awareness.**   For some participants, the abuse they received was attributed to behaviours and other markers related to the pandemic, rather than their perceived ethnicity. Many participants described adopting practices, in particular mask-wearing, from China/East Asia prior to the pandemic reaching the UK. By choosing to wear face masks when this was not yet commonplace in the UK, this identified them as targets for abuse.

*"I had to go to the surgery, the clinic. Because I've been worried, that's why I have to put the face mask on. As I walk into the clinic, two adults they come at me, calling me name as 'coronavirus'." (Participant 9)*

*"some of Chinese kids, students, they have more knowledge or more concern about this because what happened in China in Wuhan and they start wearing masks in cities and they get spat on. . ." (Participant 8)*

One participant speculated that face mask use triggered a perception that the wearer could be harbouring infection.

*"if we're kind of wearing masks on the street whilst walking, not for exercise, and most people aren't wearing masks whereas they see you wearing a mask they might kind of feel suspicious as if you have been infected or as if you have a problem." (Participant 1)*

That face masks were readily adopted despite the potential of serving as a symbol to draw stigma suggests that risk of the virus was perceived to be greater than risk of abuse, or that participants felt a responsibility to wear masks despite the negative attention this might bring. These attitudes were informed in part by the legacy of SARS in 2003.

## Pandemic legacies

**Legacies of previous pandemics.**   Despite recognition that face mask use might serve to identify them as a target for abuse, many participants had a strong belief that face masks were an effective and important measure against COVID-19, and that wearing face masks was the 'natural' thing for individuals to do to protect themselves and society generally.

*"if I have the disease I'll stop spreading it to other people, if I don't have the disease I stop catching it and, if everybody wearing it I think the chance to get [COVID-19] is very, very minimal." (Participant 8)*

*"its natural, it's something natural that they would do, just put on their mask. Not just to protect them, it's to protect the others" (Participant 6)*

*"I do this without the government guidance because of my age and I know face masks will help me." (Participant 11)*

The experience of previous respiratory disease outbreaks, namely SARS, was highlighted as a key factor underlying the willingness of people to undertake proactive measures to contain COVID-19, such as adopting face masks.

*"Because of the SARS in 2003. . .Quite a lot of people died during that year so when coronavirus came people started to get alert, ok wear masks." (Participant 14)*

*"having had previous experience with SARS. . .I think they don't want a repeat of what's happened so I think people were probably a bit more compliant with lockdown and they were quite clear in the message that they were sending out to people" (Participant 5)*

Indeed, belief in the importance of face mask use was associated with incredulity that many people in the UK chose not to wear them, extending as far as one participant being encouraged to return to China for their own safety.

*"[my parents] think all the people will go out without wearing face mask, they think this behaviour will be very dangerous and my parents even talk us maybe if possible all of you can come back to China." (Participant 1)*

This perception of the UK from China directly contrasts with the perception of China from the UK at that time and implicates the role of different media sources in shaping attitudes and beliefs around the pandemic. Additionally, the sense of individual responsibility may also have more broadly reflected the contrasting societal attitudes to the pandemic between the UK and China and Hong Kong at this time.

**Ascription of blame.** Participants described apportioning of blame for the pandemic, through negative media reporting towards China, that was believed to have contributed to stigmatisation. Political leaders were criticised for propagating this and exacerbating ill-feeling towards the Chinese community.

*"It doesn't help when you're here, see the BBC reporting all the time when they introduce it saying that the virus where it started off from China." (Participant 4)*

*"there was a lot of media around Trump saying it was the 'Chinese virus', I feel like that hasn't really been great for the Chinese community." (Participant 5)*

When contemplating the future, many participants expressed concern that implication of blame the COVID-19 pandemic would have a longlasting negative legacy for the UK-Chinese community, in terms of persistent discrimination.

*"it will take time for people to kind of reintegrate again. . .for people to kind of not look at the Chinese community as kind of like the people who spread the virus or who brought the virus over. . ." (Participant 5)*

*"Will they still be able to access care or anything without discrimination?. . .and it's quite scary 'cos imagine if you're an elderly UK-Chinese person and you were so vulnerable. . .I mean how would that affect your care?" (Participant 7)*

In this way, the attribution of stigma was reinforced by the apportioning of blame for the pandemic in public discourse upon China and, by extension, and whether intentionally or not, individuals of Chinese heritage. This in turn fed into participants concerns of the long-lasting legacy from the pandemic.

## Individual versus societal responses

**Extent of freedom.** All participants felt that the government in the UK was more liberal than in mainland China and Hong Kong, and detailed more stringent measures in China/East Asia to contain the virus. This contrasted with the UK response, which many participants felt was comparatively 'more vague'.

*"the funeral was at the end of month and he [uncle] had to go back to Hong Kong for a 14 days quarantine. . . and he's been tagged as well. He can't leave the flat for 14 days.. . .They know where he is." (Participant 4)*

*"I think the [UK] response was a bit more vague and a bit more kind of like it's up to you to do your bit rather than you need to stay indoors, you can't go out, kind of, yeah, leadership really." (Participant 5)*

Contrasting governmental styles were perceived as reflecting other societal differences between the UK and in mainland China. People in the UK were perceived as less willing to adhere to measures such as wearing face masks precisely because the UK is more 'democratic'.

*"the [British] people they have a lot of freedom so they don't follow the government" (Participant 3)*

*". . .the Chinese government has more power and authority over the decisions whereas here it's more democratic so I suppose the decision-making is more difficult to decide and enforce the restrictions" (Participant 1)*

While participants described the differences in the approaches between the UK and China, they interestingly did not explicitly identify one as better than the other.

**Implicit faith in government.** Despite the contrasting natures of measures imposed, most participants expressed faith in both the UK and Chinese/Hong Kongese governments in handling the pandemic. This faith translated to a belief that people should, and would, follow whatever the government dictated.

*"we have to just accept or respect and also follow their [UK government] command or decision and do whatever they tell us to do." (Participant 3)*

*"During the time of crisis, everyone's listening to the government, everyone wants to do their part and everyone is. . . we're all trusting our governments to help us through this" (Participant 7)*

The implicit faith in government also extended towards the apparatus of the state to support UK-Chinese individuals who encountered xenophobia. This was demonstrated in the section

described previously when participants supported those who had been victims of xenophobia to report it to the police. This faith persisted despite the challenges participants faced in navigating a media environment that exacerbated stigmatisation.

## Discussion

To our knowledge, this is the first qualitative study to describe the experiences of UK-Chinese people during the COVID-19 pandemic. Participants reported widespread stigma and xenophobia, and this was attributed in part to the adoption of practices that were legacies of the SARS outbreak in 2003, specifically wearing face masks. Participants described negative media representations feeding into the attribution of blame and stigmatisation. Importantly, participants drew parallels between government styles which underpinned a belief about the willingness of societies to adopt measures to limit spread of SARS-CoV-2, a willingness that was partly influenced by the legacy of SARS.

The xenophobia and stigmatisation reported in our study is similar to that described in observational studies of people of East-Asian heritage in the USA during the COVID-19 pandemic [6–9]. In one quantitative survey of 235 participants, 34% reported experiencing discrimination, with 14% reporting being threatened or harassed [7]. These findings were echoed in cross-sectional surveys of 218 Bhutanese and Myanma refugees in the USA, nearly one-third of whom experienced discrimination related to the pandemic. In another observational study of over 500 Chinese-American families, over half reported being directly targeted by COVID-19-related racial discrimination [9]. Beyond this, participants in a qualitative study of healthcare workers in the USA and Canada identifying as Asian reported experiencing, and perceiving, 'racial microaggressions' early during the pandemic which included verbal and physical abuse similar in nature to that reported by participants of our study [21].

Many mechanisms may underlie the exacerbation of racism and xenophobia during the COVID-19 pandemic, and these experiences must be considered in the context of legacies of sinophobia and orientalism in Western Europe and the USA [22]. Sociologists have proposed that, while racism and xenophobia are not 'natural' reactions to the threat of a virus, they have arisen in the US through historical legacies that view those of East-Asian descent as 'foreign and presenting a higher risk of transmission of disease' [6]. This implicates the 'behavioural immune system', which describes psychological responses to perceived pathogen threats [23], and has been associated with anti-immigrant sentiment generally; either through an evolved predisposition against unfamiliar others as potential carriers of infection, or as a negative by-product of hypervigilance against everything unfamiliar [24]. This system predicts that people exposed to more disease express more negative attitudes to groups identified as 'other'. Supporting this, an analysis of over 400000 US respondents to an online survey assessing implicit and explicit racial attitudes suggested that higher general local disease rates predicted greater racial prejudice among both White and Black respondents [25]. Likewise, an analysis of the stigmatisation of individuals of Chinese ethnicity in Canada during the SARS outbreak in 2003 emphasised the rationalisation of cultural racism at this time, with the generalised avoidance of those stigmatised on the basis of individual self-protection, and the framing of SARS as a consequence of globalisation [4]. Studies examining racial prejudice in the context of COVID-19 rates may provide further insight into the drivers of racism and xenophobia experienced by people of Chinese ethnicity during the pandemic.

The importance of face masks was a key theme identified, with participants contrasting practices in mainland China and Hong Kong with those in the UK. Participants attributed the ready adoption of masks to the experience of SARS and belief in masks being effective, despite absence of empirical evidence to support this at the time. Masks also served as a physical

symbol through which stigma could be attributed: masks have been specifically cited in the verbal abuse directed at people of Chinese ethnicity [21, 26–28]. Participants indicated that, early in the UK pandemic, mask-wearing was widely adopted by UK-Chinese individuals, despite recognition that it could serve to precipitate xenophobic abuse. A mixed-methods study assessing face mask use among 91 students from Hong Kong studying in the UK early in the pandemic found that over two-thirds felt stressed when wearing masks in public due to perceived stigmatisation [14]. This stigmatisation shares similarities with that experienced by people whose dress emphasises their identity as 'other'. For example, hijab-wearing Muslim women [29] and turban-wearing Sikh men [30] report greater levels of discrimination than their counterparts in survey studies from the US. Our study provides additional insight supporting these findings in the context of COVID-19, attributing racism and stigmatisation to external markers of pandemic awareness such as face masks, rather than solely perceived 'ethnic phenotype'. Indeed, instances of xenophobia described by Asian healthcare workers in North America included references to the use of protective measures such as gloves and face masks [21]. This contrasts with the conclusions of a qualitative study of the experiences of individuals of predominantly Mongolian ethnicity in Delhi, India, during the COVID-19 pandemic, which described widespread racism and stigmatisation experienced by this community, but attributed this to 'racial hypervisibility', i.e. the physical appearance of this group [31]. It is challenging to reconcile these findings given the markedly different national contexts, and further qualitative studies in diverse settings, and including individuals of Asian and non-Asian ethnicity, may provide further insight into the extent to which these external markers drive, or potentiate, stigmatisation. It is worth noting, however, that xenophobia and stigmatisation associated with face masks extends beyond their use. One study of people of Chinese ethnicity living in France reported that, as Europe became the epicentre of the pandemic and masks became mandatory, Chinese migrants were accused of having "wrapped up all the masks and sent them back to China" [27]. It is unclear if this experience was mirrored in the UK, however our interviews were almost entirely conducted prior to face masks becoming mandatory. Future interview studies may provide insight into how stigma around mask-wearing changed as their use became normalised among the general population.

Participants described media representations of China as apportioning blame and contributing to xenophobia. People of Chinese ethnicity interviewed in France explicitly linked terminology such as 'yellow alert' in the media to attacks and discrimination [27]. Additionally, use of the term 'Chinese virus' increased tenfold on Twitter following its first use by Donald Trump on March 16[th] 2020 [32], and experiences of racism on social media in the USA was associated with worse wellbeing and depressive symptomology among 209 respondents of a survey of individuals self-identifying as Asian [33]. These findings echo the accounts of our study participants, and suggest that language expressed in traditional and social media, and by political leaders, can shape and drive xenophobic discourse. Knowledge of this particular issue was highlighted by a 2015 World Health Organisation report in the best practices for naming new human infectious diseases [34], which advocated avoidance of geographical or cultural references precisely to avoid this attribution of blame and stigmatisation. Reporting guidelines could be developed to reduce harm caused by media coverage of pandemics in future. For example, a survey study of 300 adults in the USA demonstrated that content-targeted and audience-focused countermeasures to anti-China conspiracy theories related to the pandemic could limit, or even reverse, radicalisation [35]. Even beyond exacerbating negative perceptions of individuals of Chinese ethnicity, commentaries have highlighted that the overly-simplistic framing of democracy versus authoritarianism in the discourse comparing responses to the pandemic in the UK (and the USA and Western Europe generally) and mainland China, may have inhibited knowledge sharing and cooperation that could have ameliorated the global impact of the pandemic [22, 36, 37].

The measures taken in mainland China to control the pandemic, while effective, were described as 'draconian' in a qualitative study of rural villagers in Hunan, China [38]. As a semi-autonomous region within China, the initial response in Hong Kong diverged, and was less restrictive, with face mask use not becoming mandatory until July 2020 [39]. However, a cross-sectional survey of behaviours in Hong Kong in the very early stages of the pandemic described widespread voluntary adoption of stringent infection control measures, such as isolation and face mask use [13]. Indeed, a study incorporating ethnographic fieldwork and interviews in five Chinese localities highlighted the active role of rural and urban residents in the pandemic response, rather than simply passive receivers of state directives [37]. The willingness to adopt and accept stringent measures was attributed, in one study, to a 'culture of moral compliance' rooted in traditional values of respect for authority and conforming to public moral standards [40]. This view is supported by the study of villagers in Hunan, who adopted 'radical and disproportionate' lockdown measures through a community-based response rather than top-down coercion [38]. Likewise, the predominantly urban respondents to a survey in Hubei province indicated strong support for quarantine, and punitive action against non-compliant individuals [41]. Although these values cannot be directly ascribed to UK-Chinese people, we found a consistent theme of implicit faith in government among our interviewees. This was associated with willingness to comply with restrictions and belief that others would follow the rules, and may partly explain why participants did not suggest which governmental approach to managing the pandemic was preferred. This contrasts with findings from a study involving a general sample of UK residents who, despite being personally willing to adhere to restrictions, described lack of trust in government or faith in others to comply [42]. Our findings suggest that the 'culture of moral compliance' may persist in the UK-Chinese diaspora, despite the more 'liberal' governmental approach and lack of reciprocation from other UK residents.

## Strengths and limitations

To our knowledge, this is the first qualitative study to rigorously explore the experiences of people of Chinese ethnicity in the UK with regards to the COVID-19 pandemic. Indeed, there are few qualitative studies exploring the experiences of individuals of Chinese or Asian ethnicity outside of East Asia during either the COVID-19 or SARS pandemics. Participants were diverse in terms of age, sex, place of birth, year of emigration to the UK (if applicable) and education level. Our sample was also broadly reflective of the UK Chinese population, the vast majority of whom are born outside of the UK [43]. Conducting the study as the pandemic emerged meant that we were able to investigate peoples' lived experiences in real-time, rather than relying on recall of events.

Our study does however have some important limitations. This was a study of UK-Chinese individuals. The UK-Chinese population is a heterogenous group comprising individuals who self-identify as being of Chinese ethnicity, and includes individuals who are of Chinese heritage born in the UK, and individuals of Chinese heritage born outside the UK who have migrated to the UK. The majority of people (8/16) in our sample were born in Hong Kong; with 4/16 born in Mainland China. Although we did not identify differences in responses between people in each group, findings may not transfer to Chinese populations in other countries. Interviews were conducted over the telephone, which may have affected rapport between the participants and interviewer and prevented non-verbal cues from being interpreted. However, telephone interviews are recognised to yield rich data and may be better suited to collecting data on sensitive topics due to a perceived sense of anonymity [44]. Our study was conducted early in the pandemic, and as such only reveals insight into experiences

and attitudes of UK-Chinese individuals in the UK at that time. Follow-up studies would be informative in identifying how perspectives may have developed and changed, and whether the stigmatisation experienced has been sustained.

## Conclusions

This study contributes new understanding of the lived experiences of the UK-Chinese community during the COVID-19 pandemic. We report widespread experiences of xenophobia and racism, in part driven by the early adoption of infection transmission reduction measures such as face masks. These measures were adopted through insight into the previous SARS pandemic and practices in mainland China/East Asia, however they identified this community as targets for abuse. Media representations of China and COVID-19 were also felt to contribute to stigmatisation, and there was widespread concern for a longlasting negative legacy on the UK-Chinese community.

COVID-19 is the first pandemic to cause significant worldwide disruption in the information age, where individuals can obtain real-time information from friends, family, and media sources. The experiences of this pandemic will guide responses to future infectious disease outbreaks. We describe considerable stigmatisation and xenophobia experienced by the UK-Chinese community and explore factors exacerbating this. Awareness of this should be increased at all societal levels, to guide policy and safeguard minority communities from stigma, social and economic harm in future.

## Supporting information

**S1 File. Interview topic guide.**
(DOCX)

**S2 File. Consolidated Criteria for Reporting Qualitative Studies (COREQ) checklist.**
(PDF)

**S1 Table. Table of illustrative quotes.**
(DOCX)

## Acknowledgments

The views expressed in this publication are those of the authors and not necessarily those of NIHR, NHS, the Wellcome Trust, or the UK Department of Health and Social Care. For the purpose of Open Access, the author has applied a CC BY public copyright licence to any Author Accepted Manuscript version arising from this submission.

## Author Contributions

**Conceptualization:** Pippa K. Bailey, Katie Wong.

**Data curation:** Mohammed Al-Talib, Katie Wong.

**Formal analysis:** Mohammed Al-Talib, Qiaoling Zhou.

**Funding acquisition:** Katie Wong.

**Investigation:** Mohammed Al-Talib, Katie Wong.

**Methodology:** Mohammed Al-Talib, Pippa K. Bailey, Katie Wong.

**Project administration:** Katie Wong.

**Supervision:** Pippa K. Bailey.

**Validation:** Mohammed Al-Talib, Qiaoling Zhou, Katie Wong.

**Writing – original draft:** Mohammed Al-Talib.

**Writing – review & editing:** Mohammed Al-Talib, Pippa K. Bailey, Qiaoling Zhou, Katie Wong.

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
