## [Decision Letter · Decision Letter 0]

2 Aug 2022

PONE-D-22-14344The experiences of UK-Chinese individuals during the COVID-19 pandemic: a qualitative studyPLOS ONE

Dear Dr. Al-Talib,

Thank you for submitting your manuscript to PLOS ONE. After careful consideration, we feel that it has merit but does not fully meet PLOS ONE’s publication criteria as it currently stands. Therefore, we invite you to submit a revised version of the manuscript that addresses the points raised during the review process.

It is very hard to find additional reviewer for this paper, but the reviewer 1 has some very constructive comments for improving this paper. I hope the author will find them useful. 

We look forward to receiving your revised manuscript.

Kind regards,

Tianyang Liu

Academic Editor

PLOS ONE

Journal Requirements:

Reviewers' comments:

Reviewer's Responses to Questions

**Comments to the Author**

1. Is the manuscript technically sound, and do the data support the conclusions?

Reviewer #1: Partly

2. Has the statistical analysis been performed appropriately and rigorously? 

Reviewer #1: N/A

3. Have the authors made all data underlying the findings in their manuscript fully available?

Reviewer #1: No

4. Is the manuscript presented in an intelligible fashion and written in standard English?

Reviewer #1: Yes

5. Review Comments to the Author

Reviewer #1: Thank you for the opportunity to review the article “The experiences of UK-Chinese individuals during the COVID-19 pandemic: A qualitative study”. The paper aims to investigate the lived experiences of Chinese individuals living in the UK during the COVID-19 pandemic. I have enjoyed reading some of the empirical findings but think substantial improvements are needed.

1. Research aim/design: I find the research aim “to investigate the lived experiences” a bit too broad and vague. There are too many aspects of lived experiences even when specified to COVID. Reflecting this, the interview questions (S1) are also rather general and unfocused. What was the underlying rationale of asking these questions? Why are the ‘family and overseas’ and ‘the future’ relevant? What did the authors want to find in the first place?

2. Significance/study contribution: Although little research has been conducted to understand “the experiences of UK-Chinese individuals in the UK early in the pandemic”, there is a large literature on anti-Chinese/Asian sentiment and racism in general. And earlier studies have also investigated this issue during previous pandemics such as SARS. How does this study add any new insight to these two strands of research?

3. Sampling: The authors have relied on a very small sample which overwhelmingly consisted of people who were born in Hong Kong. To what extent are the findings from this sample generalisable to the UK-Chinese population? I am very suspicious about whether people who were born in Hong Kong and people who were born in Mainland China share the same feelings and attitudes especially when it comes to their family and relatives’ experiences in their home country (Hong Kong vs Mainland China). The inclusion of only 4 individuals who were born in Mainland China seems very problematic.

4. Findings/analysis: Overall, I find the findings scattered, which is linked to my previous comment on the lack of focus in research aim/design. Some of the findings are interesting (e.g., the participants did not explicitly show preference between the British and Chinese governments; an implicit faith in government) but very limited in-depth insight has been generated.

6. PLOS authors have the option to publish the peer review history of their article (what does this mean?). If published, this will include your full peer review and any attached files.

Reviewer #1: No

---

## [Author Response · Author response to Decision Letter 0]

5 Sep 2022

ACADEMIC EDITORS COMMENTS

Authors’ response: Thank you. We have checked our manuscript to ensure it meets PLOS ONE’s style requirements. 

Authors’ response: Thank you. We have updated the methods section regarding ethical approval so it now reads “Ethical approval was granted from the University of Bristol Faculty of Health Sciences Research Ethics Committee (reference number 94685).” We have also amended the methods section regarding provision of consent and this now reads: “All participants provided informed verbal consent to participation which was audio-recorded and transcribed.”.

Authors’ response: Thank you for pointing out this error. We have updated the funding information section to match the financial disclosure, and also corrected the ordering. To clarify the funding, Katie Wong received a grant from the Elizabeth Blackwell Institute, University of Bristol to conduct this research. This grant was derived from a Wellcome Trust block grant to the Elizabeth Blackwell Institute. 

Authors’ response: Thank you. We have now uploaded anonymised interview transcripts to the University of Bristol’s Research Data Repository. Access to these data is controlled, given the personal and sensitive nature of topics discussed and the fact that we cannot rule out the risk of identification, even after anonymisation. Individual researchers will need to gain approval to access to the controlled data through the University of Bristol via the Data Access Committee (DAC), before data can be shared after their host institution has signed a Data Access Agreement. We have provided a full explanation of how to access the data, including relevant URLs and DOIs, in the revised cover letter. We have also provided this in the Data Availability section of the online submission system.

 

Reviewers' comments:

Reviewer 1

Reviewer #1: Thank you for the opportunity to review the article “The experiences of UK-Chinese individuals during the COVID-19 pandemic: A qualitative study”. The paper aims to investigate the lived experiences of Chinese individuals living in the UK during the COVID-19 pandemic. I have enjoyed reading some of the empirical findings but think substantial improvements are needed.

Author response: Thank you for taking the time to review our manuscript. We are happy to hear you enjoyed reading some of the findings. We appreciate your suggestions to improve our manuscript. We believe that our amendments detailed below address the concerns you have raised and our manuscript has been strengthened as a result. 

1. Research aim/design: I find the research aim “to investigate the lived experiences” a bit too broad and vague. There are too many aspects of lived experiences even when specified to COVID. Reflecting this, the interview questions (S1) are also rather general and unfocused. What was the underlying rationale of asking these questions? Why are the ‘family and overseas’ and ‘the future’ relevant? What did the authors want to find in the first place?

Authors’ response: Thank you for these comments. We agree that the statement in our abstract/introduction that we aimed to “investigate the lived experiences” is too broad and vague. Our primary aim was to explore the impact of the pandemic on the UK-Chinese community, specifically to find out how perceived cultural and ethnic identity impacted experiences of the pandemic, and how these experiences compared to those of family or friends in China/East Asia. 

We were also interested to explore the extent to which behaviours and practices were influenced by practices in Mainland China. As such, the ‘Family and overseas’ section was relevant to explore how UK-Chinese individuals received information from China/East Asia, their perceptions of the different responses (both societally and governmentally) in the context of the pandemic and the extent to which this shaped their behaviours in the UK. Family and relatives experiences in China/East Asia acted as comparators to the experiences of the study participants. Likewise, the ‘Future’ topic investigated participants’ perspectives on the lasting impact of the pandemic. We have clarified and expanded on the study aims in the abstract:

“Infectious disease outbreaks have historically been associated with stigmatisation towards minority groups, specifically those associated with the geographical region that the disease was first identified. We aimed to investigate how the emerging COVID-19 pandemic was experienced by UK-resident individuals of Chinese ethnicity: how their perceived cultural and ethnic identity influenced their experiences, and how early insights into the pandemic in China influenced attitudes and behaviours.”

We have also restated the aims of our study in the introduction to now read:

“We aimed to investigate i) how perceived cultural and ethnic identity impacted the experiences of UK-Chinese individuals during the pandemic, ii) how behaviours and practices of UK-Chinese individuals were influenced by insight into the pandemic responses in China/East Asia, and iii) the perspectives of UK-Chinese individuals on the long-term impacts of the pandemic on the UK-Chinese community.” 

We agree the questions presented in the topic guide are general. The guide was designed to list the broad topics to be covered during the interview, and to guide the interviewer in their questioning. The questions listed were not asked verbatim to participants. A long and detailed guide was not considered practical to use in these interviews. The guide had to allow flexibility in specific question construction, order, and timing. We have updated the methods section to state this which now reads:

“A flexible topic guide was developed by the study team (S1 File). This served as a broad guide of topics to discuss: the questions listed were not asked verbatim to participants and evolved as interviews progressed.”

2. Significance/study contribution: Although little research has been conducted to understand “the experiences of UK-Chinese individuals in the UK early in the pandemic”, there is a large literature on anti-Chinese/Asian sentiment and racism in general. And earlier studies have also investigated this issue during previous pandemics such as SARS. How does this study add any new insight to these two strands of research?

Authors’ response: Thank you for this comment. We have edited our manuscript to better highlight the new insights provided by our research, and place our work in the context of the existing literature on anti-Chinese/Asian sentiment. Since the submission of our manuscript, some further relevant papers have been published which we have also discussed and referenced in our manuscript.

The new insights provided by our study are summarised:

a) the experiences of people of Chinese ethnicity in the UK/Europe in the context of the COVID-19 pandemic have been poorly described, with most of this research stemming from the USA/American continent. Although there may be commonalities, the experiences are context and individual specific, and so the experiences of US Asians will have limited transferability to a UK-Chinese population

b) we provide an in-depth description of participants’ experiences in their own words. There is very little rigorous qualitative research that has explored the experiences of people of Chinese/Asian heritage living outside of Asia during the COVID-19 pandemic. The vast majority of existing research, most of which stems from the USA as mentioned above, is limited to quantitative surveys. Much of this existing literature we have now referenced in our manuscript. However the information derived from surveys is considerably less rich or in-depth than that collected through qualitative research. We have identified only 3 relevant qualitative studies, although in notably different contexts, which we have also referenced in our manuscript, the most relevant of which is a study of experiences of healthcare workers of Asian heritage in North America [Reference doi:10.9778/cmajo.20210090].

c) our research was undertaken in a country essentially unaffected by SARS, but severely impacted by COVID-19. There were only 4 UK cases of SARS, so while the experience in other countries affected significantly by both infections may have been similar, the UK did not have this experience. The largest outbreak outside Asia was in Canada. The most relevant study is cited as reference 3 in our manuscript [Reference doi:10.3201/eid1002.030750], however this study incorporated a range of measures to gauge anti-Asian sentiment in the USA including reviewing newspapers and internet sites, group discussions and analysing phone calls to a Centre for Disease Control telephone service. What interviews that took place were described as “informal”, and there was no formal analysis of these. As such, we feel that the strand of research into anti-Asian sentiment in the context of SARS is limited, and lacks qualitative insight.

d) the timing of our research study provided contemporaneous insight into the experiences of this group as the pandemic emerged, and compared the experiences of UK-Chinese individuals with those of family members and contacts in China/East Asia, and compared governmental responses, in real time. 

f) with regards to what this study adds to existing research around anti-Chinese/Asian sentiment generally, a key insight is that participants in this study described racism and stigmatisation related to infection control behaviours in response to the pandemic (e.g. face masks), as well as physical features/perceived ethnicity. Related to this, it offers new and we believe interesting insight into how decisions regarding adopting of these practices were made – and how this was balanced against the risk of stigmatisation. We have also now included some discussion of our findings in the historical context of orientalism and sinophobia, and the role of the media in perpetuating this.

We have made substantial changes to our discussion, and expanded our conclusions, to address the points you have raised and hope that the new insights our study provides are now more clear. 

3. Sampling: The authors have relied on a very small sample which overwhelmingly consisted of people who were born in Hong Kong. To what extent are the findings from this sample generalisable to the UK-Chinese population? I am very suspicious about whether people who were born in Hong Kong and people who were born in Mainland China share the same feelings and attitudes especially when it comes to their family and relatives’ experiences in their home country (Hong Kong vs Mainland China). The inclusion of only 4 individuals who were born in Mainland China seems very problematic.

Authors’ response: Thank you for these comments.

We aimed to achieve a diverse sample, rather than a population representative sample which is associated with quantitative research. Qualitative studies seek to provide in-depth and diverse insights into a phenomenon and do not attempt to generalise the results to a population of interest [Reference https://pubmed.ncbi.nlm.nih.gov/10891968/]. Most qualitative studies are necessarily small scale to provide depth and contextualised detailed information, and thus the criterion of generalisability (external validity) cannot be applied. Instead, transferability is the extent to which the concepts and theories are relevant to other settings. Researchers can compare their results with studies conducted in different healthcare contexts, regions, or populations; position their findings with other theoretical frameworks; and describe the study setting and participant characteristics in detail so readers can judge the transferability of the findings to their own context [Reference https://journals.lww.com/transplantjournal/Fulltext/2016/04000/Qualitative_Research_in_Transplantation__Ensuring.9.aspx]. 

In our study sample, we achieved diversity with respect to sex, age, educational level, place of birth and year of migration to the UK. We have provided participant details to allow readers to judge the transferability of the findings to other individuals in other contexts. We recognise that findings may have limited transferability to other Chinese populations, and this was not an aim. We have added a statement to the limitations section stating that findings from this study may not transfer to a population of Chinese individuals from other countries.

The UK-Chinese population is a heterogenous group comprising individuals who self-identify as being of Chinese ethnicity, and includes individuals who are of Chinese heritage born in the UK, and individuals of Chinese heritage born outside the UK who have migrated to the UK. The most recent UK census states that only 23.7% of people of Chinese ethnicity were born in the UK [Reference https://www.ethnicity-facts-figures.service.gov.uk/summaries/chinese-ethnic-group] which is just above the proportion of UK-born participants in our study. Hong Kong was until 1997 a UK territory, and the majority of emigrees of Chinese ethnicity to the UK are from Hong Kong. The inclusion of fewer individuals born in Mainland China reflects the UK-Chinese population as a whole.

We acknowledge that governmental responses in Mainland China and Hong Kong need to be distinguished, and your concern that individuals born in Mainland China and Hong Kong may not share similar feelings or attitudes with regards to their families’ experiences. The measures initially imposed in Hong Kong were less restrictive than in mainland China, but there were still strict quarantine and isolation requirements earlier than when introduced in the UK. However, there is primary research from Hong Kong that describes how individuals there adopted more stringent measures – more in line with the control measures in Mainland China - out of choice. For example, mask-wearing in Hong Kong was near universal in January 2020 despite it not becoming mandatory until July 2020 [Reference doi:10.12809/hkmj209015; Reference doi:10.1016/s2468-2667(20)30090-6]. We have edited our manuscript to state to the reader that responses in China and Hong Kong differed. This section reads:

 “The measures taken in mainland China to control the pandemic, while effective, were described as ‘draconian’ in a qualitative study of rural villagers in Hunan, China [35]. As a semi-autonomous region within China, the initial response in Hong Kong diverged, and was less restrictive, with face mask use not becoming mandatory until July 2020 [36]. However, a cross-sectional survey of behaviours in Hong Kong in the very early stages of the pandemic described widespread voluntary adoption of stringent infection control measures, such as isolation and face mask use [13]. 

4. Findings/analysis: Overall, I find the findings scattered, which is linked to my previous comment on the lack of focus in research aim/design. Some of the findings are interesting (e.g., the participants did not explicitly show preference between the British and Chinese governments; an implicit faith in government) but very limited in-depth insight has been generated.

Authors’ response: Thank you. On review of our manuscript, and in light of your previous comments, we feel that the findings seemed scattered in large part because our aims were too broadly stated and unfocussed. By stating our aims more clearly, as above, we hope this has been partly addressed. We have also made our analysis more focussed by restating and condensing some of the themes, and amending the discussion to tie in with our stated aims more closely. 

We are happy to hear that you found some of the findings interesting, and believe that the amendments we have made in light of your comments have highlighted some of our other findings. We have more clearly stated the new insights generated, and included greater discussion of how our research adds to existing literature around anti-Asian/anti-Chinese racism generally, and in the context of infectious disease pandemics, including SARS and COVID-19. Importantly, we have highlighted the importance of describing the negative experiences of this community, and factors that may have exacerbated this, such that lessons can be learned in future infectious disease outbreaks.

---

## [Decision Letter · Decision Letter 1]

1 Dec 2022

PONE-D-22-14344R1The experiences of UK-Chinese individuals during the COVID-19 pandemic: a qualitative interview studyPLOS ONE

Dear Dr. Al-Talib,

Thank you for submitting your manuscript to PLOS ONE. After careful consideration, we feel that it has merit but does not fully meet PLOS ONE’s publication criteria as it currently stands. Therefore, we invite you to submit a revised version of the manuscript that addresses the points raised during the review process.

Editor's comment to the author: 

The reviews outcome from the two reviewers of your paper are conflicting. One of the reviewers--a new reviewer for your paper--is very negative, but the comments he made is a bit general. 

Although those negative comments are not very specific, I believe that you should address seriously the issues raised by the reviewer, if you want to proceed this review process. 

I also reviewed the paper by myself, and I still think it is an interesting work, and bringing new understanding for this topic. In your manuscript, you discussed that the UK-Chinese individuals experienced and perceived widespread xenophobia, in the context of media representations that ascribed blame and exacerbated stigmatisation. This in turn marked UK-Chinese individuals as targets for abuse. Participants described media representations of China as apportioning blame and contributing to  xenophobia.  For the media representation of China in western society and how this has contributed to forms of violence, I suggest to include the following paper: 

‘Can Debunked Conspiracy Theories Change Radicalized Views? Evidence from Racial Prejudice and Anti-China Sentiment Amid the COVID-19 Pandemic’, *Journal of Chinese Political Science*, DOI: 10.1007/s11366-022-09832-0

In your discussion, you mentioned that some of your findings are supported by a study of villagers in Hunan and a survey in Hubei province. Although these values cannot be directly ascribed to UK-Chinese people, you found a consistent theme of implicit faith in government among your interviewees.  I think it will be helpful to recognize and consider the differences of implicit faith between urban and rural societies in China, and between different rural communities in China. For this, it may be helpful to read the following paper: 

‘Fragmented Restriction, Fractured Resonances: Grassroots Responses to Covid-19’,* Critical Asian Studies*. 52(4): 494-511 **

We look forward to receiving your revised manuscript.

Kind regards,

Tianyang Liu

Academic Editor

PLOS ONE

Reviewers' comments:

Reviewer's Responses to Questions

**Comments to the Author**

1. If the authors have adequately addressed your comments raised in a previous round of review and you feel that this manuscript is now acceptable for publication, you may indicate that here to bypass the “Comments to the Author” section, enter your conflict of interest statement in the “Confidential to Editor” section, and submit your "Accept" recommendation.

Reviewer #1: All comments have been addressed

Reviewer #2: (No Response)

2. Is the manuscript technically sound, and do the data support the conclusions?

Reviewer #1: Yes

Reviewer #2: Partly

3. Has the statistical analysis been performed appropriately and rigorously? 

Reviewer #1: N/A

Reviewer #2: No

4. Have the authors made all data underlying the findings in their manuscript fully available?

Reviewer #1: Yes

Reviewer #2: Yes

5. Is the manuscript presented in an intelligible fashion and written in standard English?

Reviewer #1: Yes

Reviewer #2: Yes

6. Review Comments to the Author

Reviewer #1: (No Response)

Reviewer #2: I have read the article entitled The experiences of UK-Chinese individuals during the COVID-19 pandemic: a qualitative interview study and I find that the subject, although it is pertinent, the development of the article does not have sufficient quality. It is not clear why such a limited population group is interviewed. When using the qualitative method, it is necessary to have data saturation. Likewise, the categories of analysis are not supported in a strong literature review, necessary to carry out the discussion and conclusions.

The important thing about the literature review is that it is the basis for dialogue with the findings. However, in this section the authors limit themselves to a description without greater depth of the responses of the participants and leave aside the rigorous analysis.

On the other hand, in the bibliography the authors refer to 41 documents, however in the document there are no more than 20.

In conclusion, I consider that the article does not have the requirements to be published in such a prestigious journal.

7. PLOS authors have the option to publish the peer review history of their article (what does this mean?). If published, this will include your full peer review and any attached files.

Reviewer #1: No

Reviewer #2: No

---

## [Author Response · Author response to Decision Letter 1]

19 Dec 2022

Editor's comment to the author: 

The reviews outcome from the two reviewers of your paper are conflicting. One of the reviewers--a new reviewer for your paper--is very negative, but the comments he made is a bit general. 

Although those negative comments are not very specific, I believe that you should address seriously the issues raised by the reviewer, if you want to proceed this review process. 

I also reviewed the paper by myself, and I still think it is an interesting work, and bringing new understanding for this topic. In your manuscript, you discussed that the UK-Chinese individuals experienced and perceived widespread xenophobia, in the context of media representations that ascribed blame and exacerbated stigmatisation. This in turn marked UK-Chinese individuals as targets for abuse. Participants described media representations of China as apportioning blame and contributing to  xenophobia.  For the media representation of China in western society and how this has contributed to forms of violence, I suggest to include the following paper: 

‘Can Debunked Conspiracy Theories Change Radicalized Views? Evidence from Racial Prejudice and Anti-China Sentiment Amid the COVID-19 Pandemic’, Journal of Chinese Political Science, DOI: 10.1007/s11366-022-09832-0

In your discussion, you mentioned that some of your findings are supported by a study of villagers in Hunan and a survey in Hubei province. Although these values cannot be directly ascribed to UK-Chinese people, you found a consistent theme of implicit faith in government among your interviewees.  I think it will be helpful to recognize and consider the differences of implicit faith between urban and rural societies in China, and between different rural communities in China. For this, it may be helpful to read the following paper: 

‘Fragmented Restriction, Fractured Resonances: Grassroots Responses to Covid-19’, Critical Asian Studies. 52(4): 494-511 

Authors response: Thank you for these suggestions. We have reviewed these papers and have referenced them in our discussion. 

Reviewer comments

Reviewer 1

Authors’ response: We’re pleased that Reviewer 1 feels our responses to their comments are acceptable, and recommended that the manuscript be accepted.

Reviewer 2

I have read the article entitled The experiences of UK-Chinese individuals during the COVID-19 pandemic: a qualitative interview study and I find that the subject, although it is pertinent, the development of the article does not have sufficient quality. It is not clear why such a limited population group is interviewed. 

Authors’ response:

Our sample was diverse with respect to age, gender, educational attainment and place of birth/emigration to the UK. Sampling was undertaken during the COVID-19 pandemic and lockdowns in the UK. We acknowledge the possible limitations of convenience and snowball sampling in our discussion section: “Our sampling strategy may have limited the diversity of participants of the study, although participants were diverse in terms of age, educational background and gender.” 

When using the qualitative method, it is necessary to have data saturation. 

Authors’ response: As indicated in our methods we used Braun and Clarke’s method of thematic analysis. Braun and Clarke state that the concept of data saturation is not consistent with the values and assumptions of reflexive thematic analysis: V Braun and V Clarke. To saturate or not to saturate? Questioning data saturation as a useful concept for thematic analysis and sample-size rationales. Qualitative Research in Sport, Exercise and Health 2021;13(2): 201-16. https://www.tandfonline.com/doi/full/10.1080/2159676X.2019.1704846. 

Braun and Clarke suggest that other concepts such as ‘information power’ offer a more useful way of assessing data samples. Malterud, Siersma, and Guassora (2016) conclude that the more relevant information a sample holds, the fewer participants are needed. We have added the following sentence to our manuscript: ‘We assessed that the sample delivered sufficient information power, as the information provided in the interviews was rich, highly relevant and included contemporaneous accounts of current lived experiences.’ 

Likewise, the categories of analysis are not supported in a strong literature review, necessary to carry out the discussion and conclusions. The important thing about the literature review is that it is the basis for dialogue with the findings. However, in this section the authors limit themselves to a description without greater depth of the responses of the participants and leave aside the rigorous analysis.

Authors’ response: In response to suggestions from the Editor, we have discussed two other papers from the literature and added these to the reference list. We feel our discussion is rich and considers our findings in the context of the limited existing literature. As we note in the manuscript, to our knowledge, this is the first qualitative study to rigorously explore the experiences of people of Chinese ethnicity in the UK with regards to the COVID-19 pandemic. Indeed, there is limited qualitative research exploring the experiences of individuals of Chinese or Asian ethnicity in the Western world during COVID-19 or previous infectious disease outbreaks, such as SARS in 2003. There is therefore limited directly comparable literature with which to compare findings. 

On the other hand, in the bibliography the authors refer to 41 documents, however in the document there are no more than 20.

Authors’ response: We can only assume that this comment was made in error. All 41 references in the reference list were referenced in the text.

---

## [Editor Report · Decision Letter 2]

27 Dec 2022

The experiences of UK-Chinese individuals during the COVID-19 pandemic: a qualitative interview study

PONE-D-22-14344R2

Dear Dr. Al-Talib,

We’re pleased to inform you that your manuscript has been judged scientifically suitable for publication and will be formally accepted for publication once it meets all outstanding technical requirements.

Kind regards,

Tianyang Liu

Academic Editor

PLOS ONE

---

## [Editor Report · Acceptance letter]

3 Jan 2023

PONE-D-22-14344R2 

The experiences of UK-Chinese individuals during the COVID-19 pandemic: a qualitative interview study 

Dear Dr. Al-Talib:

I'm pleased to inform you that your manuscript has been deemed suitable for publication in PLOS ONE. Congratulations! Your manuscript is now with our production department. 

Kind regards, 

on behalf of

Professor Tianyang Liu 

Academic Editor

PLOS ONE